# OpenReview forum: "Benchmarking Large Language Models as AI Research Agents"
_ICLR.cc/2024/Conference — Submitted to ICLR 2024_

### Official Review · Reviewer_CHg2 · 2023-10-31

**Soundness:** 2 fair
**Presentation:** 2 fair
**Contribution:** 2 fair
**Rating:** 3
**Confidence:** 5

**Summary:**

This paper proposes a benchmark for evaluating large language models as AI research agents. The benchmark encompasses multiple modalities and a variety of tasks. Each task consists of reading a problem description and taking actions in an environment defined by the authors. Most of the problems are taken from Kaggle, with care taken to ensure that at least some of the problems were put online recently, so LLMs haven't had a chance to train on them. The authors then evaluate GPT-4 and Claude-1 on the benchmark, as well as agentic frameworks built around these models, such as AutoGPT and LangChain-React.

**Strengths:**

The paper is original, novel and addresses a significant research question. The benchmark covers a range of tasks, and the largest API-only LLMs (GPT-4, Claude-1) are evaluated. The evaluation of agentic frameworks like AutoGPT and LangChain-React is also useful, and surprising — I expected them to be better.

**Weaknesses:**

I have concerns about the relevance of the tasks chosen with respect to "AI Research". Some of the tasks are hardly "research tasks":
- CIFAR10
- IMDB
- Kaggle House Prices
- Kaggle Spaceship Titanic

I'm skeptical any research is being done on the Kaggle House prices dataset or the Spaceship Titanic dataset.

The best possible action if pushing performance on CIFAR10 and IMDB at this point would be to load pretrained models from HuggingFace or TIMM and run it on CIFAR 10/IMDB, which is not research.

The LLM Tools section does not seem possible to evaluate quantitatively, and the relevance of `bibtext-generation` to AI research seems tenuous.

This means that 6/15 proposed tasks are not directly relevant to ml research.

I understand that the motivation was
> The tasks are chosen such
that they range in various difficulties and recency to test the generalizability of the research agent and
avoid data contamination.

But I don't see anything to constrain the actions of the agent. If the task is to improve the baseline model, can I "improve" it by sticking it as one branch of a model that uses a much larger pretrained model? Why / why not?

Right now, the problem I see is that the work is a collection of ~15 already existing tasks, with ~6 of them having arguable relevance to contemporary ML research. There needs to be additional work on top of this to make it a generally usable benchmark, such as constraining the actions of the agent, or defining each task more precisely, or adopting a principled definition of a research workflow and only including tasks based on that.

**Questions:**

See weaknesses.

---

> ### Author Response · Authors · 2023-11-13
> **Official Comment by Authors**
>
> Thanks for your valuable feedback and identifying the novelty in the work! We comment on the weakness below:
>
> > I have concerns about the relevance of the tasks chosen with respect to "AI Research"...
>
> Please see our overall response in the official comments.
>
> > But I don't see anything to constrain the actions of the agent…
>
> Yes we do not constrain the agent from importing pre-trained models if that gives better performance. And we agree in general that the agents mainly do well on older tasks like cifar10 or spaceship-titanic which the LM has seen during pretraining but lack of lot of creativity as seen in some newer and more challenging tasks in our benchmark such as CLRS and BabyLM which are actual open research problems. These older tasks were added to show the spread in the type of tasks agents are capable of doing and some promise in improving current agents for harder novel tasks. We view our work to give people a sense of the current state of agents on ML engineering tasks which involve a lot of open-ended decision making.
>
> The actions were all the actions that an actual human could have taken to solve that particular problem, and hence are open-ended. The easier tasks were added to show the capabilities of current agents as opposed to having a suite a super hard tasks where all agents have a 0 performance (aren’t able to making any meaningful progress in, such as BabyLM). Thus, the 15 tasks in the benchmark are not static and can evolve very easy as agent capabilities increase. We wanted to have tasks which had a decent spread of success rates so that current agents can be compared as opposed to all very easy or very hard tasks only which doesn’t make a lot of sense for a benchmark.
> Furthermore, our novel framework ensures that adding a task to the benchmark is very seamless (very disentangled from a lot of underlying agent code) which would encourage open-source contributions, and hence would make the benchmark more wholesome and varied in the types of task to be tested. We hope our benchmark can provide valuable foundation for benchmarking more complex types of research in future work. One could simply add the research problem they are working on in our benchmark in <10 mins but it is very unlikely that the agent will show a lot of creativity to make meaningful progress in it (and hence the need for a benchmark to track this).

---

> > ### Comment · Reviewer_CHg2 · 2023-11-23
> >
> > I have read the response in the official comments. My issue continues to be with the design and meaningfulness of the benchmark. Let's ignore the issues around whether the benchmark constitutes research or not and focus on the classical ML engineering tasks. Is the current design enough to draw any conclusions?
> >
> > There's a total of 5 tasks in the "classical" tier + "classical Kaggle" tier. First, that's a small number of evaluations. Within those three tasks, there is a huge variation between each model's performance and across the tasks. What conclusions should be drawn from this? See Fig 3. Is GPT-4 _significantly_ better at writing experiments for graph node classification (ogbn-arxiv) but mysteriously _much worse_ at tabular house price classification (house-price) compared to Claude-1? Why is IMDB so hard for all of the models? Is it that text classification is just intrinsically way harder to improve on than every other task?
> >
> > Some of these conclusions seem intuitively implausible given their specificity, but this is all we get from the current size / combination of tasks. If there was more than one task for, let's say, image classification in the "classical tier" (ideally 5+), we could draw stronger conclusions. Given that it's just image classification, why not plug in a bunch of small image classification datasets? Same for all the other tasks. Since a selling point of the proposed framework is that that adding another task to the benchmark is very easy, why such a small number of tasks in the "classical" tier?
> >
> > We also don't know what "simple baseline" agent performance is. Let's say I took an agent and just told it to increase the width / depth of each starter model for all the classical tasks. Let's say a human wrote the this "simple baseline" solution for each task as well. How well would it do? This is what I was hinting at by "constraining actions". This is important to know because it gives us a sanity check for (a) the intrinsic difficulty of each task (b) the difficulty of following a prompt in the environment. Without these sanity checks, interpreting the results from the benchmark is difficult.
> >
> > I like the idea of the benchmark and think the initial analysis is interesting, but important pieces are missing. It's hard to see what conclusions you can draw by running the benchmark currently.

---

### Official Review · Reviewer_UyZC · 2023-10-31

**Soundness:** 2 fair
**Presentation:** 2 fair
**Contribution:** 2 fair
**Rating:** 6
**Confidence:** 3

**Summary:**

This paper proposes a new benchmark, MLAgentBench, for evaluating AI research agents that can perform open-ended ML tasks given a task description and a dataset.
The benchmark provides a set of diverse and tasks, an environment where agents can interact with files and execute code, and metrics to measure the competence, reasoning, and efficiency of the agents.
This paper tests several famous LLM and presents the experimental results.

**Strengths:**

It proposes a novel and general framework for specifying and evaluating AI research agents on ML tasks.
It tests the feasibility and effectiveness of using common LLMs as AI research agents.

**Weaknesses:**

The testing scenario is simplistic and greatly differs from the demands of the real world to some extent, which limits its ability to accurately reflect the true capabilities of the model.

**Questions:**

Why is there no AutoGen test which should be much stronger than AutoGPT?
If the output and input exceed the LLM context length during the process, how can it be resolved?
Will the model being tested receive feedback from the evaluation results of the submission? Will it then continue to carry out operations to improve the results?

---

> ### Author Response · Authors · 2023-11-13
> **Official Comment by Authors**
>
> Thanks for your valuable feedback and identifying the novelty in the work! We answer the questions asked and comment on the weakness about the testing scenario being too simple below:
>
> > The testing scenario is simplistic and greatly differs from the demands of the real world to some extent, which limits its ability to accurately reflect the true capabilities of the model.
>
> Please see our general response in the official comments.
>
> > Why is there no AutoGen test which should be much stronger than AutoGPT? If the output and input exceed the LLM context length during the process, how can it be resolved? Will the model being tested receive feedback from the evaluation results of the submission? Will it then continue to carry out operations to improve the results?
>
> We thank the reviewer for pointing out several important aspects for clarification.
> We believe AutoGen was released after we finished most of the evaluation, and adding it will require more budget. We leave testing these new agent frameworks to our future work.
> When input exceeds the LLM context length, we simply declare it as terminated and only evaluate the progress done so far. We generally try resolve it by summarizing the observations that are too long with another LLM.
> In this work, we treat the evaluation results of the submission as a final testing step and therefore do not propagate this feedback to the LLM. The AI research agent would have had plenty of chances to receive performance feedback from the eval split of the dataset instead, and the agent is supposed to improve results base on this feedback.

---

> > ### Comment · Reviewer_UyZC · 2023-11-18
> >
> > Thanks a lot for the author's response which is very helpful in understanding the questions. The reviewer believes that benchmarking the single iteration result of agents is necessary, but still maintain that research should be a multi-round process. Therefore, learning from the evaluation feedback signals to the agent for further iterations carries significant evaluation value.
> >
> > For research agents, the evaluation should not only consider whether the final performance improvements. Evaluating research ideas should also play a significant role in the assessment, and this is a very challenging aspect. However, currently, this part is entirely assessed manually, which diminishes the value of the benchmark.
> >
> > Furthermore, concerning the content of B1 in the supplementary material, it's a bit of challenging to claim that the output is a research plan but more like a parameter tuning / structure searching plan to some extend. Therefore, this benchmark seems more like a benchmark for model performance optimization agents rather than a research agents in a general meaning.

---

> > > ### Author Response · Authors · 2023-11-20
> > > **Response by Authors**
> > >
> > > Thank you for your response!
> > >
> > > > The reviewer believes that benchmarking the single iteration result of agents is necessary, but still maintain that research should be a multi-round process.
> > >
> > > The way an agent performs in the environment is still multi-round: it is like an RL agent that takes an action in the environment, receives observation, and then decides what next actions are to be taken. So all the so called “model performances” are observed not by making just a single-shot change in the code but the agent is responsible of inferring from the observations and improve iteratively in a rewardless manner (which is typical of many real-life scenarios). The agent gets exactly the same observations as a human performing these tasks would see. The agent for sure could be refined to take into account additional signals but that is not the point of this benchmark, it is rather to observe what current agents (with the type of reasoning prompts the community typically uses) perform on ML tasks. So, research is still a multi-round process, please check Appendix D for an entire log of the agent interaction with the environment.
> > >
> > > > For research agents, the evaluation should not only consider whether the final performance improvements. Evaluating research ideas should also play a significant role in the assessment, and this is a very challenging aspect.
> > >
> > > Yes, we agree that evaluating research ideas (the hypothesis generation part of research) is very challenging. However, no benchmark exists even for the evaluation of ML engineering capabilities of these agents, which is an important part of research. We wanted to start off with something that is evalutable easily and concretely, make progress over there, and then go to the much harder task of showing if these agent show “scientific creativity”.
> > >
> > > > Furthermore, concerning the content of B1 in the supplementary material, it's a bit of challenging to claim that the output is a research plan but more like a parameter tuning / structure searching plan to some extend. Therefore, this benchmark seems more like a benchmark for model performance optimization agents rather than a research agents in a general meaning.
> > >
> > > The mentioned research plan is only for the cifar-10 task where the agent thinks it can improve performance but doing some parameter tuning. There are other research-heavy tasks such as BabyLM and CLRS which are open problems in the community and where we evaluate the agents as well.
> > >
> > > The following captures the essence of the benchmark well: We indeed want agents to do very hard research tasks but most current LLM based agents are simply unable to make progress there (again, see performance on BabyLM, CLRS). So instead of having 15 novel research tasks in the benchmark and all agents showing a 0 performance in each task, we wanted to show that the current capabilities of these agents is limited to ML engineering tasks only. The easier tasks are added to show some spread in the performance of the agents and hence have atleast some signal as to how to improve the agents. We could have some GPT-n in the future which could solve the entire benchmark perfectly and by that point, the benchmark would be evolved to have harder and harder tasks as and when we see increase in capabilities.
> > >
> > > So the goal is still to see if agents can do research. It just turns out that they can’t currently! So we instead focus on smaller experimental design type of tasks (ML engineering) to track progress and also keep 2-3 very hard tasks in the benchmark which will signal us to when these capabilities have started arising.

---

### Official Review · Reviewer_TJMi · 2023-11-01

**Soundness:** 2 fair
**Presentation:** 3 good
**Contribution:** 2 fair
**Rating:** 3
**Confidence:** 3

**Summary:**

This paper proposes a benchmark for AI research tasks for large language model (LLM) based agents. They collect 15 tasks from various machine learning problems, ranging from “solved” tasks like cifar10 to “unsolved” tasks like BabyLM. Tasks consist of initial code and instructions to solve or improve performance for a given problem. They design baseline agents and assess their ability using the benchmark, showing that they are able to achieve >10% performance increases for many of the tasks, with varying success depending on model used, task type, and agent design. They perform a number of studies on the types of mistakes agents make, and the efficiency of agents in terms of tokens used.

**Strengths:**

The ability of LLMs to conduct research is an interesting question that seems to follow naturally from the focus on the reasoning ability of LLMs. A benchmark to measure this ability could be a valuable contribution. This paper does a lot of work analyzing the types of errors agents make, which is also valuable for the community.

**Weaknesses:**

Overall, while this paper contains some interesting findings and raises an interesting question, I feel that it needs more polishing before it’s ready for publication.

1. Though the main point of the paper is proposing a benchmark, there is relatively little information on how these tasks were selected and how the benchmark was constructed. The benchmark consists of only 15 tasks, and while there is some detail on the attributes that made the authors select these tasks, there are not enough to convince me that this is a thorough enough benchmark in its current state. Issues such as potential data contamination, formalizing what aspects of research are being tested and how these are addressed by the chosen tasks should be explained further.
2. Relatedly, the framing of this as an AI benchmark is slightly misleading. It tests LLM ability to complete research in empirical machine learning tasks, but does not test their ability to do other types of research (e.g. more qualitative or descriptive research not based purely around achieving performance increases). Though some elements of the process may remain the same, 15 tasks from machine learning are not enough to test an agent’s overall research ability.
3. The success criteria for these tasks involves models improving performance by >10%. This strongly limits the types of tasks that can be tested, as discussed in the previous point. However, it also provides limited insight into the performance of the model. One of the strengths of this paper is the analysis of the types of errors models make, but this is all done through human evaluation. For tasks involving many reasoning steps where lots can go wrong, it would be more valuable to see some breakdown of which step the model failed at or some other type of automated error analysis.

**Questions:**

1. Did you explore how likely it is that these tasks and their solutions appeared in training data? For example, for solved tasks, it seems likely that models would have seen high performing solutions already. Did you study this at all, either by looking at the data or comparing the similarity of model output to existing solutions?
2. Was human annotation performed by the authors? What was the criteria for annotation? If it was not performed by the authors, how were annotators recruited/paid?
3. What aspects of the tasks chosen make them suitable for testing research in general? What kind of research do you expect an agent achieving perfect performance on your benchmark to achieve?

---

> ### Author Response · Authors · 2023-11-13
> **Official Comment by Authors**
>
> Thanks for your valuable feedback and recognising the usefulness of having a benchmark to keep track of the progress of LLMs to perform open-ended machine learning tasks. We have some comments about the weaknesses highlighted about the paper:
>
> > there is relatively little information on how these tasks were selected and how the benchmark was constructed.
>
> Thanks for the suggestion for clarifying how tasks are selected. As noted in section 2.4, “the tasks are chosen such that they range in various difficulties and recency to test the generalizability of the research agent and avoid data contamination”. To avoid the chance of data contamination, we selected Kaggle competitions that were launched only in the last 2-10 months at the time. We also narrowed the types of tasks to empirical machine learning tasks to specifically test the ability of AI research agents on these well-defined evaluable tasks. The tasks were hence chosen so that current agents can show some spread in performance on the benchmark.
>
> > It tests LLM ability to complete research in empirical machine learning tasks, but does not test their ability to do other types of research.
>
> Please see the overall response in the official comments. Indeed, our benchmark focuses on the problem of having research agents develop/edit ML models given a task description and a dataset, since the final result can easily evaluated. In contrast, the broader types of research are generally difficult to evaluate objectively and automatically. We leave this as an important future work direction. Please also see our further response in official comments.
>
> > It would be more valuable to see some breakdown of which step the model failed at or some other type of automated error analysis.
>
> Our success criteria of improving performance by >10% indeed only shows the broad stroke of the performance of AI research agents. However, we argue that this metric still successfully sets precedence in the evaluation of AI research agents. We have also proposed process-based evaluation, but it is one of the most challenging aspects of the agent evaluation pipeline. As mentioned in the paper, we have attempted to automate this evaluation process with LLMs such as GPT3.5, but the result is too inaccurate to be used. Thus we provide our human error analysis to shed light on the importance of such process evaluation but leave its automation to future work.
>
> To answer your questions about the work:
>
> > Did you explore how likely it is that these tasks and their solutions appeared in training data?
>
> The models we used are claude and gpt4, for which we have no information about the training data. So we try to avoid data contamination by selecting recently lauched Kaggle data (within 2-10 months) and current research benchmarks.
>
> > Was human annotation performed by the authors? What was the criteria for annotation?
>
> The error analysis is performed by ourselves. The failure modes were carefully chosen by manually inspecting agent logs while also ensuring that those error modes captured some more fundamental limitations of LLM agents. For common failure modes, we outline when is an agent run classified into that mode:
> Hallucination: We observe that the agent changes part of the code which degrades performance, but still claims that it performed the said task and takes the “Final Answer” action incorrectly.
> Debugging: We often notice that for our engineering type of tasks, being able to debug code was very important. Very often, agents would get stuck in trying to debug a piece of code it wrote and exceed the max steps permitted.
> Bad Plan: A run has a bad plan if the initial research plan of the agent itself does not help solve the goal.
> We will include these detailed descriptions in our appendix.
>
> > What aspects of the tasks chosen make them suitable for testing research in general? What kind of research do you expect an agent achieving perfect performance on your benchmark to achieve?
>
> The tasks we chose generally involve building an ML model on a given dataset for as high performance as possible (Note that >10% is an arbitrary threshold we choose to measure the baseline agents, but this threshold could be arbitrarily high instead as a more challenging scenario). Basic improvements could be simply achieved by applying existing knowledge on hyperparameter tuning, data augmentation, architecture changes etc. This allows even naive agents to start demonstrating performance early on. If an agent achieves perfect performance on our benchmark, the agent would have to be very good at ML engineering (which requires software eng skills + knowledge of a lot of state-of-the-art ML methods) to better performance. Furthermore, CLRS and BabyLM in our benchmark are actual open research problems and having an agent solve those would indicate some potential of actually being able to do “research”. To the extreme, the perfect AI research agent should be able to achieve groundbreaking research.

---

> > ### Comment · Reviewer_TJMi · 2023-11-17
> >
> > Thank you for your response. While this did clear up my questions regarding data contamination and annotation, my overall concerns about the benchmark remain. My main objection is to the idea that this benchmark tests research ability. If this is the goal of the benchmark, "research" needs to be rigorously formalized, even within the context of ML tasks, to avoid giving the wrong impression about what this benchmark tests (especially since "research" is a very broad concept).
> >
> > In my view, while these tasks represent a form of research, they're more accurately described as ML engineering tests (something that your main response seems to agree with). This is not to say that engineering can't be research, but it is by no means a comprehensive research benchmark, even for empirical ML. In its current form, this benchmark tests a very specific type of quantitative, performance-oriented machine learning research ability. As easy as it may be to add new tasks, I don't see how the evaluation procedure can be easily extended to other types of tasks (e.g. tasks without concrete performance thresholds to improve).
> >
> > You mention components of research, for example, iterating over solutions, in your main reply. However, there is nothing in this benchmark that requires LLMs to follow this process. As reviewer CHg2 pointed out, if the best solution is loading a pre-trained model rather than iterating over solutions, the model may just do this. While it could be argued that this does demonstrate a research skill (prioritizing efficiency and knowing existing work), it isn't testing the model's ability to make novel discoveries and learn from past mistakes. Again, this may be acceptable, but it has to be clearly defined in your definition of what research is and the skills that your tasks are designed to test. What qualities specifically made you choose these tasks, beyond recency & concerns about data contamination? What skills do they require a model to demonstrate? What would it mean for a research agent to generalize? The properties of an ideal research agent are not defined clearly, and this leaves too much room for interpretation.
> >
> > If an agent achieves perfect performance on your benchmark, what can we expect from it? Should we expect that it can replace a senior scientist? A PhD student? An ML engineer? How do you define research? Which facets of that definition are tested by this benchmark, and which are not? This is the kind of operationalization I would need to see---both for the definition of research and the tasks selected---to justify accepting this benchmark.

---

> > > ### Author Response · Authors · 2023-11-20
> > > **Author Response**
> > >
> > > Thank you for your response!
> > >
> > > We agree that research is way complex and multi-dimensional to be captured in a bunch of tasks. That is not even our goal, we do not want to say Agent A is a better “researcher” than Agent B if it does well on our benchmark. As opposed to some ultimate comparison, we would like to explore the **potential** of these agents to do research.
> > >
> > > The moonshot goal indeed is to have agents come up with novel ideas and performing “scientific discovery” in it’s truest sense. We started off with that goal in mind but we realised that most current LLM based agents were unable to make a lot of progress along that direction due to lack of creativity. We then realised that experimental design is something where these agents would be very helpful in a research pipeline.
> > >
> > > > If this is the goal of the benchmark, "research" needs to be rigorously formalized, even within the context of ML tasks, to avoid giving the wrong impression about what this benchmark tests (especially since "research" is a very broad concept).
> > >
> > > Hence, to answer this, we formalize “research”, in the context of this work as designing experiments, running them, making inferences and updating the next set of experiments to perform. This is an ability which is shown by current LLM-based agents (the agent logs on cifar-10 in the supplementary capture this very well). We again agree that this does not compass a lot of “broad research”, and that is fine as agents perfecting the above definition are also an integral part of the research cycle. We would be happy to clarify this in the paper: by explicitly writing what are the different aspects of research (hypothesis generation, experimental design, testing, and inference making) and what this benchmark specifically focusses on (the latter 3 of this cycle).
> > >
> > > > As easy as it may be to add new tasks, I don't see how the evaluation procedure can be easily extended to other types of tasks
> > >
> > > Yes, **we agree that this benchmark only captures the ability of agents to do empirical research where the outcome can be measured by concrete performance metrics**. We would not be able to evaluate the ability of an agent to come up with a novel theoretical proof. We would clarify this in the paper as well. This is a **tradeoff we deliberately chose**: having a much more open-ended benchmark with almost no good way of evaluating or focussing on a subset of research problems where evaluations are meaningful and progress can be tracked.
> > >
> > > > You mention components of research, for example, iterating over solutions, in your main reply. However, there is nothing in this benchmark that requires LLMs to follow this process
> > >
> > > The agents infact require iterating over solutions. The goal of improving >10% accuracy might look very trivial and that the agent would simply use some pretrained model for this. In practice, the agent makes a lot of useful inferences for this: for cifar-10 the model observes some overfitting when it expanded the model size and adds dropout in the next step, it still does not reach the threshold and then does some learning rate optimization. Again, the **agent log on cifar-10 in the supplementary would clarify a lot of questions related to the improvement aspect**.
> > >
> > > > If an agent achieves perfect performance on your benchmark, what can we expect from it? Should we expect that it can replace a senior scientist? A PhD student? An ML engineer?
> > >
> > > If an agent achieves perfect performance on our benchmark, **the agent would have to be very good at ML engineering (which requires software eng skills + knowledge of a lot of state-of-the-art ML methods) to better performance**. Furthermore, CLRS and BabyLM in our benchmark are actual open research problems, and having an agent solve those would indicate some potential of actually being able to do “research”. We would not like to specify a “profession” that these agents would be able to replace, as people have varied preferences of skillsets for each of these, and those skill set boundaries are very blurred. That being said, an ML engineer will be the closest as the current benchmark is kaggle-heavy but again, CLRS and BabyLM are very research-oriented. Here is our understanding of the different difficulties of tasks and “who” would be able to solve them perfectly.
> > >
> > > **Canonical Tasks, Classic Kaggle**: Someone having taken introductory ML/DL classes, something at the level of an **undergraduate**.
> > >
> > > **Kaggle Challenges**: A very **strong ML engineer**, who is up-to-date with a lot of current papers and techniques and is somewhat creative as in combining ideas from 2-3 papers. Again, it is hard to say if “a PhD student” would be able to do these successfully given the variety and most students are trained in a specific sub-domain.
> > >
> > > **Current Research**: Anyone who does research **(PhDs, postdocs professors at all levels)**, these are open problems in the community and were added in the benchmark as it is.

---

> > > > ### Comment · Reviewer_TJMi · 2023-11-23
> > > >
> > > > Thank you for another thorough response. While this does add clarity to what you hope this benchmark will achieve, I still believe that calling it a benchmark for ML research is misleading. We agree that full fledged scientific discovery is not tested by this benchmark, and also seem to agree that ML-engineering skills are. If this benchmark was described as testing ML-engineering capabilities, I would have significantly less complaints, as it fits with what is actually shown. CLRS and BabyLM are indeed open research problems, and I see how an agent that does well on this benchmark may be able to contribute to research tasks like these, but it is much more accurate as a whole to say that you are testing ML-engineering agents.
> > > >
> > > > Regarding iterating over solutions, I recognize that the agent tested did iterate. However, this does not mean there is something inherent to the tasks that requires them to iterate. Whether iteration is something you want to test or not goes back to the question of how research is formalized, but the same holds for any other skills going into your definition of research ability. In general, I see that providing guarantees on what tasks require may not be possible, however, describing in more detail what skills tasks are designed to test (e.g., going beyond saying you selected recent tasks and describing that X tasks test skill A, Y test skill B, etc.) would help to demonstrate that the benchmark is covering sufficiently diverse tasks.
> > > >
> > > > In its current form, I don't feel this paper is ready to be published at ICLR, so I will keep my score the same. However, I encourage the authors to continue to refine the paper as I do believe the benchmark holds value.

---

### Official Review · Reviewer_5hPW · 2023-11-04

**Soundness:** 4 excellent
**Presentation:** 4 excellent
**Contribution:** 4 excellent
**Rating:** 10
**Confidence:** 4

**Summary:**

The authors introduce a new benchmark for AI agents that conduct AI research, called MLAgentBench. The authors provide a baseline algorithm based on GPT-4 which achieves some successes on various tasks within the benchmark, but which also demonstrates the large gap between the state-of-the-art and expert performance, inspiring future research in this area. The code for the paper is open-sourced, enabling researchers to reproduce these results and build new models that push the frontier of AI research agents.

**Strengths:**

- The authors provide a novel, rigorous, open-source evaluation benchmark to galvanise and focus the community on the important topic of AI research agents.
- The benchmark includes a well-chosen range of tasks, which represent a diverse range of machine learning challenges, from standard problems to more difficult Kaggle challenges.
- The paper is very well-motivated, and the descriptions are clear and lucid throughout.
- The authors provide a state-of-the-art agent, alongside baselines and ablations. The "no-retrieval" ablation, for instance is a good choice that enables the reader to establish scientific intuition and may inspire future work.
- Qualitative analyses of agent behavior provides an insight into the capabilities and limitations of the agents.

**Weaknesses:**

- In the Results section figures, it is unclear what is meant by "baseline". Could the authors please clarify which model this is? It would be good to make this clear in the figure captions.
- It would be useful to have a slightly longer description of each of the MLAgentBench tasks in Table 1. At the very least, a few words in the caption explaining the definitions of the terms used in the "task" column (e.g. node classification, improve speed, implement tool) would be very useful.
- The future work section is too thin for a benchmark paper. Can the authors provide some more suggestions for possible future lines of agent research? For this paper to be maximally impactful they should take this opportunity to provide researchers in the field with some inspiration to drive rapid progress on this benchmark!

**Questions:**

See "Weaknesses".

---

> ### Author Response · Authors · 2023-11-13
> **Official Comment by Authors**
>
> Thanks a lot for identifying merits in our work to have a benchmark to keep track of the capabilities of Agents to perform machine learning tasks. Below, we address some of the highlighted weaknesses:
>
> > It is unclear what is meant by "baseline"
>
> As mentioned in 2.1, “The starter code mostly implements a simple baseline model that we can compare with during evaluation”. We will add this clarification to the experiment section and caption.
>
> > It would be useful to have a slightly longer description of each of the MLAgentBench tasks in Table 1
>
> We will include clarification of each type of task in the caption, i.e. node classification task refers to classifying node in a networks to different classes; improve speed task refers to improving the run time of a given script; implement tool refers to implementing an LLM based helper script such as performing literature review. We will also add a more detailed per task description in the appendix.
>
> > The future work section is too thin for a benchmark paper.
>
> Good point! We will include a separate section detailing future works including: 1) improving agent architecture, such as incorporating literature review, structured memory, better hierarchical actions and even employing multiple subagents to divide up the responsibilities; 2) incorporating more tasks from different domains, such as performing experiments in biology, chemistry, and physics; 3) improving the evaluation pipeline, such as automatic evaluation of agent reasoning process and evaluating agent in more general research scenarios.

---

> > ### Comment · Reviewer_5hPW · 2023-11-22
> >
> > I thank the authors for their response. I agree with them that this is a good starting point for a benchmark on the engineering side of AI research. While I am sympathetic to some of the concerns of the other reviewers, what stands out to me here is that this is a well-written paper with good motivation, strong open-source contribution and which provides a much-needed starting point for future work in this area. In my view, this makes it a strong paper, in spite of the tasks not yet capturing the full range of AI research possibilities. So I continue to strongly recommend acceptance here.
> >
> > A point of advice to the authors: it is very effective to make the changes you promise in an updated version of the PDF which you upload to OpenReview for the reviewers to comment on. As far as I can tell, you have not yet incorporated my feedback into the PDF. I trust that you will do so in time, but for future work, please consider updating the PDF during the rebuttal period.

---

### Author Response · Authors · 2023-11-13
**Global Response by Authors**

Thank you all for your valuable feedback! **We are glad to see all reviewers agree that our paper and framework are novel and address interesting/significant research questions. Reviewer 5hPW, TJMi, and CHg2 also highlighted that the evaluation results in our paper could be a valuable contribution.** However, there is also a common concern regarding the types of tasks we selected being different from the broad definition of research. We would like to address this below:

A common concern raised by many reviewers is that many tasks do not look research-like and are very easy. We would like to emphasize that we view research as a multi-step process: generating a hypothesis, performing experimental design, getting results and inferring from it. **In this work, we focus on testing the capabilities of these agents to perform experimental design and execution for empirical machine learning research, i.e. ML engineering tasks, where the evaluation criteria can be objective and quantitative.** The idea is to provide the agents a very human-like environment to work in – a folder where they can do file system actions, write code, run experiments, look at logs and iterate over these. We believe that benchmarking agent capabilities on such ML engineering tasks will be valuable and an important part of research as a whole (for e.g., reading a paper, implementing it, and verifying the claimed performance can be integrated in the framework we operate in).


Another concern raised by the reviewer UyZC and TJMi is regarding evaluation of agents. We want to note that evaluations are typically very hard (and multi-dimensional) for agents which can take such open-ended decisions and most prior works do not even focus a lot on principled evaluations/use human evaluations which are clearly not scalable. In this work, we wanted to be more concrete and automated about evaluations for ML engineering tasks and this is done by ensuring all agents submit a submission.csv which is processed by a task specific eval script. We also note that LLM based process evaluations are not reliable and instead present common failure modes of agents from human evaluation, and also evaluate agents based on their efficiency (number of tokens used). We agree that this still might not capture a lot of constraints for real-world applications but still **sets a precedent in the way the community should think about evaluations of agents.**

So instead of end-to-end research which is a moonshot goal, we view agents as very useful copilots that could automate a lot of engineering that goes in empirical research. This would allow scientists to try a lot of interesting hypotheses (which still they come up with) and have these agents implement the said task provided to them. **Furthermore, we have ensured that adding a task to the benchmark is very seamless** (very disentangled from a lot of underlying agent code) which would encourage open-source contributions, and hence would make the benchmark more wholesome and varied in the types of task to be tested. We hope our benchmark can provide valuable foundation for benchmarking more complex types of research in future work.

---

### Public Comment · ~Penglin_Cai1 · 2023-12-01
**A novel benchmark with great contribution**

This benchmark is an initial exploration in benchmarking LLMs as AI research agents, from which we can see the novel idea and problem setting. Although there are many imperfections, such as limited capabilities of LLMs reflected in experiments, lacking in formal formulation, the gap from the real world, and few tasks to test, these shortcomings cannot conceal the outstanding contribution of this paper. From my perspective, there is no need to be too mean to a newly designed benchmark. For instance, it is difficult to give a formal formlation of such a benchmark in which LLMs act as research agents. In conclusion, this paper opens a door to a possible world where AIs can be used for conducting research automatically, which greatly enhances productivity and reduces the load of humans. In this aspect, the great contributions of this work may be re-considered carefully.

---

### Meta-Review · Area_Chair_AjUu · 2023-12-06

**Metareview:**

The paper introduces MLAgentBench, a benchmark for AI agents to conduct AI research, particularly focusing on ML engineering tasks. The strengths include its originality, the variety of tasks, and its contribution to evaluating AI research agents. However, the benchmark faces criticism for not adequately capturing the broader spectrum of AI research, leaning heavily towards ML engineering rather than full-fledged scientific discovery. Some tasks are criticized for their limited research relevance, and there's a lack of clarity on the benchmark's construction and task selection. For example, the baselines for the involved tasks are not covered in the paper or clarified during the rebuttal.

**Justification For Why Not Higher Score:**

The decision to reject stems from the benchmark's limitations in fully capturing the essence of AI research. While it excels in testing ML engineering capabilities, it falls short in representing a comprehensive research benchmark, especially in terms of testing agents' abilities for broader, more qualitative types of research.

**Justification For Why Not Lower Score:**

Despite its limitations, the benchmark's novelty, the range of tasks it covers, and its contribution to the field of AI research agents are commendable. These aspects warrant recognition and prevent a lower score, as the paper does make a valuable contribution to the field, albeit with room for improvement.

---

### Decision · Program_Chairs · 2024-01-16

Reject